# Tourism as a Factor of Regional Development: Community Perceptions and Potential Bank Support in the Kopaonik National Park (Serbia)

**Jovana Brankov** [1,2,*], **Ivana Penjišević** [3], **Nina B. Ćurčić** [1] and **Branko Živanović** [4]

1    Geographical Institute "Jovan Cvijić" of the Serbian Academy of Sciences and Arts, Đure Jakšića 9, 11000 Belgrade, Serbia; n.curcic@gi.sanu.ac.rs
2    Institute of Sports, Tourism and Service, South Ural State University, 76 Lenin Ave., Chelyabinsk 454080, Russia
3    Department of Geography, University in Priština – Kosovska Mitrovica, Faculty of Natural Sciences and Mathematics, Lole Ribara 29, 38220 Kosovska Mitrovica, Serbia; ivana.penjisevic@pr.ac.rs
4    Belgrade Banking Academy, Faculty of Banking, Insurance and Finance, Zmaj Jovina 12, 11000 Belgrade, Serbia; branko.zivanovic@bba.edu.rs
*    Correspondence: j.brankov@gi.sanu.ac.rs

**Abstract:** This paper represents a case study examining perceptions about tourism and reactions of the local community and bank decision makers to its development. The survey method was applied to establish the community's attitude towards the impact of tourism in different spheres of life in the Kopaonik National Park (Serbia). The sample of 195 adult respondents covered inhabitants of 16 communities located within the wider area of the national park. In parallel, the potential support of banks for tourism development was examined on a sample of 21 banks. The survey results identified strong positive attitudes towards tourism and the presence of tourists among the local population. Compared to other categories, the community's members employed in tourism had more favorable perceptions of tourism. The findings of the study also revealed that younger and better educated members of the population had more positive attitudes towards tourism impacts. Certain independent economic variables (the impact of tourism on job creation) and non-economic ones (the impact of tourism on activities of the community, reactions to the presence of tourists) significantly predicted the community´s support for tourism. An analysis of potential bank support showed that future community involvement in the tourism industry should be initiated by an adequate approach and credit policy instruments in the wider area of the Kopaonik National park.

**Keywords:** residents' perceptions; tourism impacts; bank support; Kopaonik National Park; Serbia

## 1. Introduction

A balanced and harmonious relationship between tourists and the community, as well as between them and organizations providing tourist services, constitutes the basis for the successful development of tourism [1]. For this reason, academic attention in recent decades has largely been focused on the social impact of tourism and understanding the local community's perception of this industry. As tourism directly affects the community's quality of life, scholars have agreed that local self-governments, planners and entrepreneurs, as well as all other subjects involved in the development of tourism, must take into account the opinions and desires of the local population if they seek to achieve long-term sustainability [2–6].

Academic literature emphasizes the positive consequences of tourism for communities due to its potential for job creation, expansion of investment, infrastructure development and general

improvement of well-being, as has been found in many host areas [7,8]. On the other hand, the local community can view this industry in a negative light on account of environmental damage and harmful socio-cultural influences. This situation has also been found in many local areas [9,10]. Usually, residents will be conscious of the dual implications of tourism, and their perceptions are likely to be affected by balancing the trade-offs between what they consider to be benefits and the costs of those benefits. Accordingly, the term "tourism development dilemma" is used in the academic literature to express how the community reacts to the development of tourism, namely the acceptance of its benefits, but also awareness of its potential negative consequences [11]. Some authors have suggested that such acceptance is of key importance for the successful development of tourism. For example, Andriotis and Vaughan [12] (p. 172) observe that "the balance of residents' perceptions of the costs and benefits of tourism is a major factor in tourist satisfaction and is therefore vital for success of the tourism industry".

A unique dimension of the relationship between tourism and the community is present in protected areas, given the existence of more pronounced environmental concerns. In the case of these areas, academic authors are unanimous in believing that finding ways to deal with community worries and incorporate them into the decision-making process is crucial for the long-term sustainability of protected areas and for maximizing benefits to the local population [13–15]. If protected areas are to benefit from the implementation of conservation-based management actions, the community needs to support them. Distressed members of the community will express opposition to regulations related to protected areas, refuse to participate in actions taken by the authorities and hesitate to cooperate with other stakeholders, directly eroding the privileged status of the areas in question [16]. In certain cases, the inclusion of a territory into the system of protected areas also involves the resettlement of local residents, which sometimes exposes them to the risk of impoverishment and makes them hostile as a consequence [17]. A proper understanding of perceptions and desires of the local population can prevent such circumstances and produce reasonable policy guidelines and management procedures, as well as reduce negative impacts [14,18].

In contrast to this prevailing attitude, new theories have been put forward in recent years arguing that local support is not necessarily required for the survival of protected areas and that conservation can be imposed despite the local community's opposition. These theories rely on the fact that the local population often belongs to poor and marginalized rural communities, whose opposition to conservation (and the possible development of tourism), although persistent, may remain entirely ineffective against the driving forces behind protected areas. In such cases, scholars argue that disregarding the interests of local groups does not result in a long-term threat to the safety of protected assets [19].

Factors that affect community support for the development of tourism have been extensively investigated by various scholars. These factors, including community attachment, community concern and an eco-centric attitude [20,21], place attachment [22], personal attitudes [23] and perceived benefits [24], may strongly influence local residents' support. In addition, previous research often examined the connection between support for the development of sustainable tourism and its observed effects. Nevertheless, relatively few studies have been carried out on community support for tourism among residents of the Balkan countries, specifically those living on the territory of protected areas.

Due to centralization of the management of protected areas in Serbia in the past, the significance of local populations living on the territory of these areas or along its boundaries has been minimized [25]. Today, a specific situation is present in the Kopaonik National Park, which has a dominant role in the tourism of Serbia's national parks and in the mountain tourism of Serbia. Starting from the 1980s, winter tourism progressed rapidly on Kopaonik Mountain and nowadays it is the leading human activity in the wider area that is characterized by significant comparative advantage in this industry [26]. Concurrently, due to outstanding natural values, the area including the ski resort was declared a national park in 1981. The conflicts between tourism and active protection have grown over time, since the highest parts of the mountain represent the largest ski resort in the country and,

at the same time, a protected natural area. A large number of entities, starting from investors and private companies with accommodation and service facilities and including the public enterprise managing the national park, participate in organization and presentation of the tourist offer. In such circumstances, an important challenge is to find an active role for the local population and incorporate this group into the development of tourism.

In spite of the great importance of tourism to the Kopaonik area, little is known about the local community's perceptions towards the development of tourism. Moreover, no research has been carried out to examine the rural residents' reactions or perceptions towards visitors and the influence of tourism as a whole. To fill these gaps, our research aims to identify community perceptions and specific factors that influence support for the tourism industry. The study of Demirović et al. [27] served as a kind of antecedent of this research. However, that study was mainly focused on examining perceptions about tourism impacts on socio-ecological systems and covered a wider area than in the case of the present research.

Besides constructs that refer to residents' perceptions about tourism impacts, another important issue that has been neglected by investigators is the question of the potential investments in tourism that would affect the well-being of the community. It should be emphasized in particular that no research has been devoted to the Kopaonik area´s attractiveness for potential investments by banks that would be beneficial to the local population and would serve to start small businesses in the tourism sector. Accordingly, one of the goals of the present research was to fill this gap by determining the area´s attractiveness for potential lending by banks, with particular emphasis on those types of lending that would be stimulating to the community.

Being aware of residents' attitudes towards the influence of tourism would allow scholars and users to better understand the status of tourism in this research area. Such information might be useful when creating development strategies and policies which would be reconciled with the needs of the local community and ensure its support. Furthermore, making real opportunities for tourism lending in this area known to prospective investors is essential as a means of creating financial incentives, reducing the regional disparities of development [28] and guaranteeing future community involvement in the tourism industry.

## 2. Theoretical Background and Research Hypothesis

The significance of the relationship between host communities and visitors has been underlined by different researchers who claimed that the effective development of tourism is strongly conditioned by support provided by the community [29].

In recent decades, there has been a growing number of studies on residents' perceptions of the impact of tourism due to the fact that many community leaders, local governing structures and governments are concerned by the active opposition to tourism from the host communities [6,30,31]. In order to explain how perceptions about the impact of tourism are created, most of these studies were conducted within the framework of specific theories.

As the most relevant theory related to residents' perceptions about the impacts of tourism, as well as the conceptual framework for the present research, the social exchange theory (SET) has been used to assess support of the host population for the development of tourism. This theory has been the framework most commonly accepted in explaining the reaction of inhabitants to tourism, since it permits the identifying of heterogeneous views based on experience and psychological factors [32]. The theory assumes that every interaction provides an exchange of resources between individuals and groups [33]. It is therefore important to identify the exchange of tangible or intangible resources that hosts and visitors may offer and receive in this process. When applied to attitudes, if the host residents perceive that they are likely to benefit from such exchanges without unacceptable costs, they will participate in the process of exchange with tourists and support the community-based development of tourism. On the other hand, if they feel that expansion of tourism would produce more costs than benefits, they are likely to oppose this development [15,34–37]. Based on social exchange principles,

numerous studies worldwide have detected an important link between observed benefits to the host population and support for tourism activities [38–41]. At the same time, it was found that the perceived costs negatively affect this kind of community support [38,39,42]. Obviously, the perceived benefits and costs are important predictors of residents' attitudes.

Previous studies indicate that there are three main elements involved in the process of exchange in the development of tourism, namely, economic, sociocultural and environmental impacts [43,44]—all of which must be taken into account in order to ensure an effective evaluation of its sustainability. Such an evaluation can be achieved using the triple bottom line approach to impacts, commonly employed in studies dealing with the sustainable development of tourism [32,45]. Some scholars additionally investigate legal impacts as a specific dimension of the process of exchange in the development of tourism [15,46,47].

It is accepted that tourism can have both favorable and unfavorable impacts on the local community with respect to each of these dimensions of exchange. As far as economic impacts are concerned, tourism can provide more job opportunities and improve infrastructure [22,44], but may increase the cost of living [48]. Through the development of tourism, cultural exchange and preservation of local culture are enhanced [21,49], but other effects may be less welcomed, such as changes in social and family structure and emergence of cultural practices adapted to suit the needs of tourists [50]. Often, tourism is considered responsible for environmental pollution and noise [51], but its activities can encourage the community to participate in maintaining the local environment, deepening awareness of the need for environmental protection [52]. Tourism can also stimulate various economic, socio-cultural and environmental issues, such as crowds and prostitution [53], crime [54], overutilization of community heritage [55], etc. In summary, past studies have confirmed that the more positively residents perceive the impacts of each of the three domains of tourism exchange, the more prepared they are to support the development of this industry. On the other hand, if they perceive more negative impacts from tourism, their support will be reduced. Therefore, the following hypothesis is presented:

**Hypothesis 1 (H1).** *Residents' positive perceptions of tourism impacts positively affect their support for the sustainable development of tourism.*

An evaluation of the positive and negative effects of tourism by residents may also be moderated by certain factors affecting the intensity of perception. These moderating variables can include the sociodemographic profile of the inhabitants, the residents' personal characteristics, economic dependence on tourism, the place of residence, the feeling of attachment to the community, etc. Scholars have drawn various conclusions from an analysis of the impact of sociodemographic factors on perceptions of the community towards tourism. While some studies concluded that sociodemographic factors do not have a causal influence [5,56] many others found significant relationships between them [57–60]. Jackson and Inbakaran [61] even proposed a sociodemographic profile of the community member who shows the most favorable attitude towards tourism development.

With respect to the factor of economic dependence, the most frequently confirmed hypothesis proposes that the greater the economic dependence on tourism, the more favorable the attitude towards tourism [62]. Working in tourism (and being economically dependent on this industry) leads to a strong positive attitude towards tourism [61]. However, the conclusions of some researchers [5,63] indicated that residents with an economic reliance on tourism not only had a strong positive attitude towards tourism, but also recognized its negative impacts, a circumstance which leads to the manifestation of strong negative attitudes as well. In light of this discussion, it is suggested that:

**Hypothesis 2 (H2).** *A resident's perceptions of the impacts of tourism are significantly moderated by sociodemographic factors affecting the intensity of perception.*

**Hypothesis 3 (H3).** *Personal benefits from tourism development are positively related to perceived positive tourism impacts.*

Scholars have broadly recognized that tourism contributes to regional development and job creation, and so the financing of investments in the tourism sector has been widely investigated in the literature [64–66]. In the case of rural areas, the economic development achieved through tourism is often synonymous with small-business promotion, and this industry is heavily characterized by small, family-centered enterprises [65]. It has been confirmed that, in addition to their own funding sources, bank landing represents a significant source of capital for business start-up of numerous small entrepreneurs [67,68]. The attitudes of commercial banks towards potential lending to small tourism-based enterprises has also been investigated in studies where it was pointed out that these attitudes are dictated by many factors, for example the development of the tourism market, the amount of information available to banks, professionalism of the entrepreneurs, etc. [69]. Regarding this issue, Fleischer and Felsenstein [65] stated that "even minimal support can yield substantial economic and social returns". In light of the foregoing considerations, the following hypothesis is presented:

**Hypothesis 4 (H4).** *Banks respond positively to potential tourism lending due to a developed tourism market.*

## 3. Research Method

### 3.1. Study Location

The Kopaonik National Park is situated in the southern part of central Serbia on Kopaonik Mountain, one of the largest (2758 km$^2$) and highest (2017 m) mountains in Serbia [70–72]. Recognized as an area of exceptional natural value, it was established in 1981 and today it includes a total area of 11,969.04 ha, covering the highest and most valuable part of the Kopaonik massif. Three levels of protection were established within the national park, with the third level covering the largest part, 57.68%, the second level covering 29.94% and the first and most demanding level covering 12.38% of the park´s total area [73]. The area of the national park is mountainous, the highest point being the peak Pančićev Vrh, with an elevation of 2017 m, and the lowest point at approximately 640 m above sea level [74]. Various geological, geomorphological and pedological features and specific climatic conditions caused the formation of rich and diverse life, especially vegetation, with the result that Kopaonik Mountain is today considered a center of the arctic-alpine flora and floristically the richest mountain on the Balkan Peninsula. With a huge number of high-mountain endemics, Kopaonik Mountain together with the Stara Planina and Suva Planina Mountains represents the second most important center of biodiversity of endemic flora in Serbia, after the Prokletije and Šar Planina Mountains [75].

The territory of Kopaonik National Park is located in the Raška and Brus municipalities, within 16 cadastral municipalities, viz., Kopaonik, Crna Glava, Jošanička Banja, Kremiće, Tiodže, Semeteš, Badanj, Lisina, Bozoljin, Brzeće, Ravnište, Kneževo, Gočmanci, Livađe, Paljevštica and Kriva Reka. A protection zone is established around the national park´s territory, and it covers a total area of 20,538.27 ha—of which, 10,379.23 ha is within the Raška municipality, 5945.17 ha is within the Brus municipality and 4213.87 ha is within the Leposavić municipality [73]. Management of the national park is entrusted to the "National Park Kopaonik" public enterprise, funded in 1993 [76]. The enterprise´s main tasks are protection and improvement of the environment, and all activities are conducted in accordance with relevant legal acts (Figure 1).

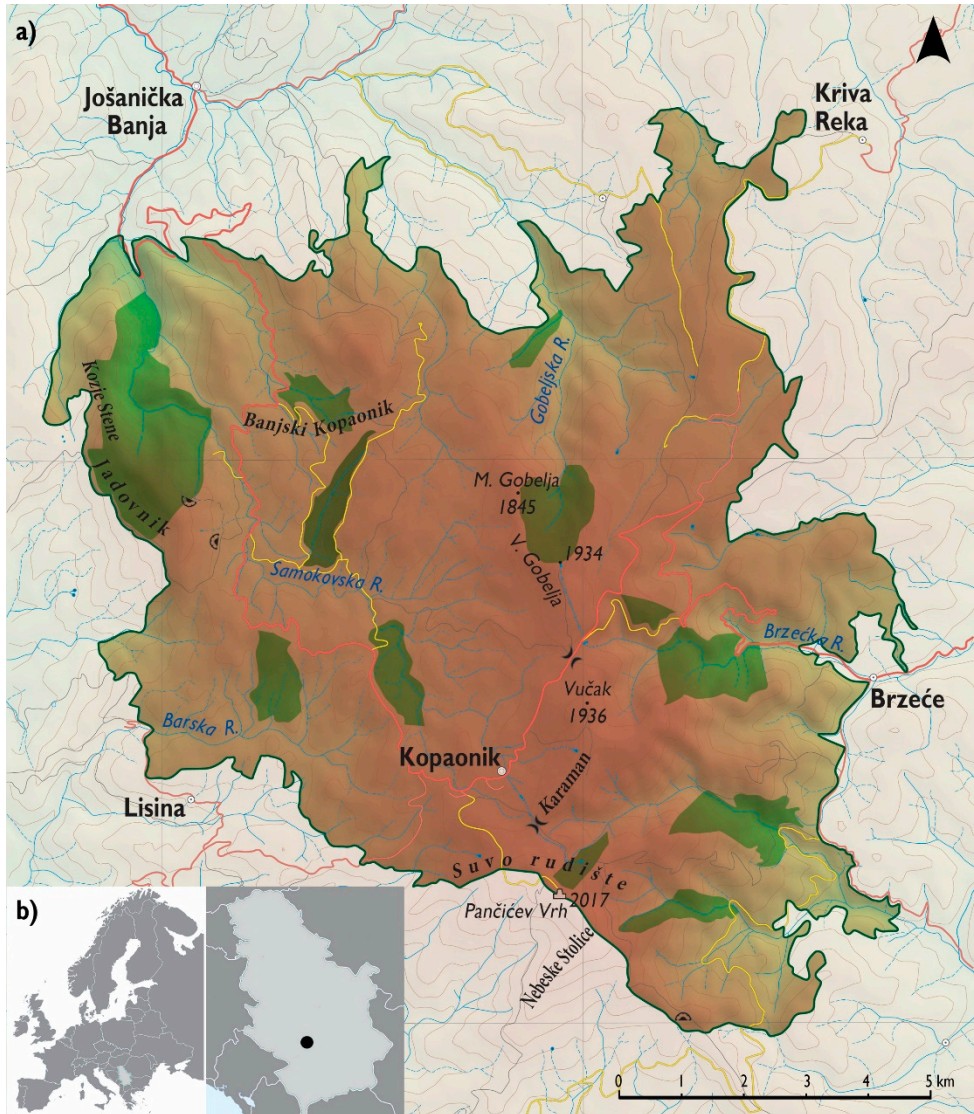

**Figure 1.** Location of the study area: (**a**) Kopaonik National Park (green areas represent the first level of protection); (**b**) position of the Kopaonik National Park in Serbia and Europe.

Besides being acknowledged as a region of significant natural value, this area is also known as the oldest and largest ski resort in Serbia and the second largest ski center in Southeast Europe, after Bansko in Bulgaria [77]. The development of ski tourism started in the 1930s, when the first organized skiing course was conducted, and today the ski resort has approximately 62 km of prepared paths and ski trails, with an artificial snow-making system covering approximately 97% of the entire territory of the ski resort. Winter tourism has been developing rapidly, especially from the 1980s, and so there is growing conflict between sustainability of natural resources on the one hand, and economic development on the other. In the period 1971–2017 in Kopaonik (settlement), the number of tourists per year increased from 4692 to 131,178, and the number of nights spent per year increased from 22,016 to 550,962 in 2017 [78,79]. In addition to tourism, a dominant activity is forestry. State forests are managed by the "National Park Kopaonik" public enterprise within four managing units: Samokovska Reka, Barska Reka, Gobeljska Reka and Brzećka Reka following corresponding laws. The "National Park Kopaonik" public enterprise is funded from the budget of the Republic of Serbia and from its own resources. According to financial reports, the public enterprise in the period 2009–2011 was funded mainly from its own activities, such as forestry (65%–70%) and tourism (25%—from fees charged for protected area use) and less from the budget [80].

Municipalities on the territory of the Kopaonik National Park suffered a decline in economic activities in the previous period and consequently were assigned the status of extremely underdeveloped local self-government units with a development rate below 60% of the national average [81]. Previous research suggested that the Kopaonik National Park and its protection zone can be classified as neglected mountain areas, according to criteria of the regional classification of rural areas in the EU [82]. There are numerous economic problems and structural weaknesses that have affected this situation: peripheral position in relation to urban centers; a high share of agricultural production in relation to other activities; extremely low population density; unfavorable age, educational and economic structure of the population; poor traffic connections; etc. In general, rapid urbanization in the whole country resulted in the marginalization of the village and moving the focus to urban areas [83].

Settlements in the area of Kopaonik National Park are predominantly rural, in the mountainous altitude zone (at elevations of more than 600 m above sea level) and with unfavorable size and morphological structure. During the 1980s this area was affected by an earthquake, which affected the quality of life of the communities [84]. The region is dominated by small (100–300 inhabitants) and very small (under 100 inhabitants) rural settlements with a population that is existentially dependent on agricultural production. The communities are insufficiently connected to the tourist center in Kopaonik National Park in terms of their involvement in tourist activities, as well as with the region´s main municipal centers (Raška, Brus) regarding the use of public services (healthcare, social services) and other amenities. Although the major tourist center was formed in the highest parts of the mountain, certain other settlements eventually received a specific tourist function, such settlements including Brzeće, Jošanička Banja and the illegally built settlement of Lisina. With the development of tertiary activities and some public services, the established tourist centers have assumed the functions of secondary municipal centers. However, their functional connection to the main municipal centers is still inadequate, and the equipment of their communal infrastructure is not aligned with the regimes of the protection of natural resources [82].

*3.2. Questionnaire Development*

Based on a review of the literature dealing with community-based tourism, a survey instrument was developed for this research. The local questionnaire model recommended by the UN World Tourism Organization [85] was used in designing the survey. This questionnaire included items that measured various impacts of tourism on the local community, in addition to ones addressing the control of tourism development and seeking the respondents´ overall opinion about this industry. The same methodology was successfully used in the case of other protected areas in Serbia [15] and communities in their vicinity.

The questionnaires were administered using a direct face-to-face survey methodology, which could be why this approach attained high response rates. The authors delivered the questionnaires personally to households and the interviewer invited the person opening the door to participate in the study (any respondent who was 18 years old or older could participate in the survey). Because of the moderate illiteracy present among older members of the population in these communities, and since the local residents were generally unfamiliar with survey procedures, questions were asked directly to respondents and recorded by the interviewer. The purpose of the research was explained to community members, and they were motivated to give their sincere judgement. Only one member from each household took part in the research, due to the fact that members of the same family often have similar perceptions [86]. In total, 195 usable questionnaires were collected.

The final questionnaire consisted of two main sections, which can be described as follows. The first section of the questionnaire was intended to measure residents' perceptions about various impacts of tourism. A 13 item scale was developed to measure the influences of tourism on the host residents. The categorization of items was in agreement with findings of previous studies [43,87] that indicate the existence of three major factors involved in the tourism exchange process, namely economic,

socio-cultural and environmental impacts. Some other scholars add a fourth sphere of influence to the equation—legal and moral impacts [15,47].

Four measurement items were used to capture perceived economic impact (creating new jobs for the local population; employing the young local population; increasing property value; infrastructure), while three items evaluated socio-cultural impacts (establishing new services; development of the local culture; activities of the local population). Two items measured perceived legal and moral issues (crime; decline in moral standards), and environmental impacts were investigated using four variables (environment; noise and crowds; access to sightseeing spots; utilization of natural resources). Responses in the first section were scored on a five-point Likert scale (bipolar scale, ranging from 1 = significantly worsening to 5 = significantly improving (stimulating)).

For perceptions about the control of tourism development and clarification of the overall attitude towards tourism, an eight-item scale was constructed to assess the community's opinion on these issues and the intention of its members to support the sustainable development of tourism. Four items measured attitudes towards the control of tourism development (control of tourism development by the local community; extent to which money spent by tourists remains within the community; access to places used by tourists; availability of information about sustainable tourism) and another four were related to the overall attitude towards tourism (support for the sustainable development of tourism; correct understanding of the impact of tourism on the community; attitude towards the presence of tourists in the community; overall opinion on tourism in the future). The responses in this section were scored on a five-point Likert scale (a scale ranging from 1 = strongly disagree to 5 = strongly agree). In order to ascertain the scale's reliability, Cronbach's alpha score was calculated (0.815) and an acceptable level of internal consistency for items measuring the same construct was confirmed.

The process of collecting the field data involved a stratified random-sampling method. In consultation with local community authorities, 16 communities located within the national park or its protection zone or else situated along the major traffic routes leading to this protected area were selected for this study (Table 1). Due to the large number of internal migrations in the Kopaonik area from the mountainous to the lower parts of municipalities in recent decades [88], settlements located on the territory of the national park or its protection zone were almost emptied, and so it was necessary to include low-altitude communities located close to major roads leading to this area in the analysis. The vast majority of communities covered by the survey are located in the municipality of Raška, while a smaller number of them belong to the municipality of Brus.

**Table 1.** Respondents in communities of the wider area of the Kopaonik National Park [88].

| Settlement | Municipality | Completed Questionnaires | Inhabitants |
|---|---|---|---|
| Šipačina | Raška | 5 | 126 |
| Lisina | Raška | 6 | 30 |
| Kopaonik | Raška | 16 | 19 |
| Jošanička Banja | Raška | 28 | 1036 |
| Rakovac | Raška | 10 | 173 |
| Semeteš | Raška | 17 | 90 |
| Badanj | Raška | 3 | 74 |
| Crna Glava | Raška | 7 | 203 |
| Brvenik | Raška | 3 | 64 |
| Rudnica | Raška | 16 | 334 |
| Rvati | Raška | 19 | 620 |
| Biljanovac | Raška | 10 | 533 |
| Novo Selo | Raška | 14 | 281 |
| Raška (municipality center) | Raška | 18 | 6590 |
| Kriva Reka | Brus | 4 | 384 |
| Brus (municipality center) | Brus | 19 | 4636 |

To create profiles of the respondents, demographic variables such as gender, age, education level and employment in the tourism industry were included in the research. Various research techniques were used for data analysis. The descriptive statistics, bivariate analysis and multivariate analysis (t-tests, ANOVA, multiple regressions) were performed using SPSS 20.0 for Windows in order to recognize factors that affect the particular outcome [89,90].

In the part of the research related to the analysis of the attitudes of commercial banks, the specific survey instrument was developed and carried out on the sample of 21 commercial banks (out of 26 banks operating in the system). In this way, the sample covered 80.77% of the total number of banks in the national banking system. The questionnaires were sent electronically to the addresses of banks, and the executives took part in the survey. In addition, interviews were conducted with ten executives of the banks to explain in detail the attitudes offered in the survey. The questionnaire was intended to measure opinions regarding potential lending to various spheres of the tourism industry in the Kopaonik region. The first group of questions concerned the general attitudes regarding lending to this area, as well as the selection of the most suitable activities for lending. The second group of questions was about potential lending to the "National Park Kopaonik" public enterprise.

## 4. Results

In order to identify perceptions of the community regarding the influence of tourism in different spheres, 13 tourism impact variables were defined and classified into four categories (economic, social and cultural, legal and moral, and environmental). (Table 2). Various items related to the local population's control of tourism development, as well as to correct understanding of the impact of tourism and the overall attitude towards this industry, were also included in the analysis (Table 3).

**Table 2.** Perceptions of the community about the impacts of tourism.

| Variables | Mean | SD |
|---|---|---|
| Economic Issues | 4.0 | |
| Creating new jobs for the local population | 4.1 | 0.8 |
| Employing the young local population | 4.0 | 0.8 |
| Increasing property value | 3.6 | 1.1 |
| Infrastructure (roads, water supply system, sewage, waste management, etc.) | 4.1 | 0.7 |
| Social and Cultural Issues | 3.6 | |
| Establishing new services (healthcare, communal, etc.) | 3.8 | 0.9 |
| Development of the local culture without compromising the integrity and authenticity of the community | 3.3 | 1.0 |
| Activities of the local population | 3.6 | 1.1 |
| Legal and Moral Issues | 2.4 | |
| Crime | 2.4 | 1.2 |
| Decline in moral standards | 2.4 | 1.1 |
| Environmental Issues | 3.1 | |
| Environment | 2.6 | 1.4 |
| Noise and crowd | 3.5 | 1.3 |
| Access to sightseeing spots in national park | 3.4 | 1.0 |
| Utilization of natural resources needed by the local population (fish, water, etc.) | 3.1 | 1.1 |
| 13 Impact Variables | 3.6 | |

"What is the impact of tourism in your municipality/community on the following activities? Scale: 1 – Significantly worsening; 2 – Worsening; 3 – None (makes no difference); 4 – Improving (stimulating); 5 – Significantly improving (stimulating).

**Table 3.** Perceptions about the control of tourism development and the overall opinion about tourism.

| Variables | Mean | SD |
|---|---|---|
| Local community controls tourism development (V1) | 3.3 | 1.1 |
| Money spent by tourists remains within the community (V2) | 3.0 | 1.2 |
| Access to places used by tourists (V3) | 3.5 | 1.1 |
| Availability of information on sustainable tourism (V4) | 3.2 | 1.1 |
| Support to sustainable tourism development (V5) | 4.5 | 0.7 |
| Correct understanding of tourism impact on the community (V6) | 4.0 | 0.8 |
| Attitude towards the presence of tourists in the community (V7) | 4.2 | 1.0 |
| Overall opinion on tourism in the future (V8) | 4.2 | 0.8 |

An analysis of the respondents' socio-demographic profile revealed that the male segment of the population was slightly dominant (53.8%) compared to the female segment (46.2%). The profile was marked by the following presence of different age categories: 40–49 years (22.6%), 30–39 years (19%) and 50–59 years (15.9%). As far as the education level of the local population is concerned, respondents with secondary education prevailed (46.7%), whereas the percentage of the population with a college or university degree (34.9%), as well as those with only elementary education (18.5%), was significantly smaller. In the examined sample, those who declared themselves as employed in tourism or those who had at least one or two members of the closest family employed in this industry accounted for 44.1%.

According to members of the community, tourism strongly affected it through the creation of new jobs, employment of young people and improvement of infrastructure. Among the positive impacts, residents also recognized the influence of tourism on establishing new services and activities of the local population. Community members believe that tourism does not cause any rise in crime or decline in moral standards. As for negative impacts of this industry, respondents believe that tourism has stimulated an increase in property prices. They also think that tourism has encouraged increased noise (a negative impact) and caused the creation of crowds. Half of the respondents (53.8%) believe that tourism harms the environment. As for variables pertaining to the utilization of natural resources needed by the local population and the development of the local culture, community members had a neutral opinion.

An analysis of variables related to the control of tourism development indicated that 49.7% of the population believed that the local community controls it and 41% of the population believed that the money spent by tourists remained in their community. Respondents considered that most of the sites used by tourists were accessible to the local population (61.5% of respondents gave positive answers).

The inhabitants believed that they properly understood the impact of tourism development on their community (76.4% of respondents gave positive answers), although the responses relating to the availability of information about sustainable tourism were prevailingly neutral. Mean values of the variable related to residents' opinion on the presence of tourists (4.2) suggested a strong positive perception, indicating support for tourism. The community strongly supported the development of sustainable tourism (93.3%), which indicated that certain positive effects of the prosperity of tourism on the community's overall well-being were identified. As for the scope of future tourist activities, i.e., the overall opinion about the development of tourism in the future, the local population believed that these activities should be present on a considerably larger scale.

### 4.1. Local Support for Tourism and the Influence of Socio-Demographic Variables

Bivariate and multivariate statistical analyses were used to explain the relationship between socio-demographic variables and the respondents' support for tourism and at the same time predict the level of impact of the analyzed variables on attitudes towards tourism. The t-test and one-way analysis of variance (ANOVA) were applied in order to examine the relationship between the aforementioned variables. The analysis included variables related to the control of tourism development ('The local community controls tourism development' (V1), 'Money spent by tourists remains within the community' (V2), 'The local community can easily access places used by tourists' (V3), and 'Information on sustainable tourism is easily accessible when needed' (V4)); and variables pertaining to the general attitude and support for tourism activities ('I support the development of sustainable tourism in my community' (V5), 'I believe that I understand correctly the impact of tourism on my community' (V6), 'Attitude regarding the presence of tourists in the community' (V7) and 'Overall opinion about tourism in the future' (V8)).

A series of t-tests was applied to compare the results obtained on two independent groups of people regarding the analyzed features (variables related to the control of tourism development and the overall attitude and support for this industry). It was found that women more strongly supported the claim that the local community controls tourism development (V1) (Table 4).

**Table 4.** Differences in perceptions by gender.

| Residents' Perceptions | Mean | | t-Value | $p$ * |
|---|---|---|---|---|
| | Male (n = 105) | Female (n = 90) | | |
| Local community controls tourism development (V1) | 3.16 | 3.52 | −2.26 | 0.02 |

(* $p < 0.05$).

The results confirmed that persons employed in the tourism industry (or those with at least one or two members of the closest family employed in tourism) had more positive perceptions regarding the presence of tourists in the community (V7). This group of respondents also had a more positive attitude towards the claim that the local community had easy access to places used by tourists (V3) and towards support for the sustainable development of tourism (V5). As for the overall opinion about tourism in the future, the analysis revealed a more positive attitude among respondents not employed in the tourism industry than among members of the other surveyed group (Table 5).

**Table 5.** Differences in perceptions by employment in the tourism industry.

| Residents' Perceptions | Mean | | t-Value | $p$ * |
|---|---|---|---|---|
| | Tourism Employee (n = 86) | Unemployed in Tourism (n = 109) | | |
| Attitude regarding the presence of tourists in the community (V7) | 4.42 | 4.05 | 2.61 | 0.01 |
| Access to places used by tourists (V3) | 3.81 | 3.25 | 3.51 | 0.001 |
| Support to sustainable tourism development (V5) | 4.67 | 4.38 | 2.96 | 0.003 |
| Overall opinion on tourism in the future (V8) | 4.07 | 4.38 | −2.55 | 0.01 |

(* $p < 0.05$).

A one-way analysis of variants (ANOVA) was used to compare average results of the observed attributes in specific independent categories of residents. Differences in attitudes among various age categories of respondents were identified for the variables related to the availability of information about sustainable tourism (V4) and correct understanding of the impact of tourism on the community (V6). The analysis showed that younger residents had more positive attitudes towards the availability of adequate information, i.e., in comparison with older respondents, they were more convinced that information about sustainable tourism was easily accessible. Younger respondents were also more convinced than older inhabitants that they correctly understood the impact of tourism on the community (Table 6).

**Table 6.** Differences in perceptions among various age categories of the surveyed population.

| Residents' Perceptions | Age | | | | F-Value | *p* * |
| --- | --- | --- | --- | --- | --- | --- |
| | Up to 33 | 34–45 | 46–58 | 59+ | | |
| Availability of information on sustainable tourism (V4) | 3.33 | 3.43 | 3.22 | 2.83 | 2.8 | 0.04 |
| Correct understanding of tourism impact on the community (V6) | 4.35 | 4.26 | 4.02 | 3.39 | 13.5 | 0.000 |

(* $p < 0.05$).

Similar results were obtained in the case of better educated respondents, who had more positive opinions regarding the two mentioned variables in comparison with less educated respondents. This category of respondents more strongly supported overall tourism development in the future (Table 7).

**Table 7.** Differences in perceptions among members of the surveyed population with different levels of education.

| Residents' Perceptions | Education Level | | | F-Value | *p* * |
| --- | --- | --- | --- | --- | --- |
| | Elementary School | Secondary Education | Univ/College Degree | | |
| Availability of information on sustainable tourism (V4) | 2.67 | 3.42 | 3.21 | 6.5 | 0.002 |
| Correct understanding of tourism impact on the community (V6) | 3.28 | 4.07 | 4.34 | 20.8 | 0.000 |
| Overall opinion on tourism in the future (V8) | 4.06 | 4.15 | 4.46 | 3.7 | 0.03 |

(* $p < 0.05$).

According to the obtained results, the most significant variables affecting residents' perceptions in the Kopaonik National Park are employment in the tourism industry and education. This is in accordance with the results of previous studies that emphasized the importance of the aforementioned socio-demographic factors for the population's perceptions of tourism outcomes [12,15,59]. Past studies offer several explanations of this fact. Comparison between inhabitants with high and low levels of education showed that less well-educated inhabitants believe that they are not likely to receive any offer of employment and thus will derive no benefit from tourism. Additionally, members of this category of respondents are often keen to keep the traditional way of life due to various social and cultural influences in the community.

Regarding the variable related to employment in the tourism industry, tourism workers consider that they have more knowledge about tourism impacts, a circumstance which leads to more positive perceptions. Furthermore, the economic aspect and the fact that tourism provides livelihoods to this group of respondents should not be overlooked, and so more positive opinions are to be expected. In general, all of these findings confirm basic principles of the social exchange theory—those who gain from the tourism industry recognize greater advantages than others [46].

### 4.2. Predicting the Degree of Local Community Support

A standard multiple regression analysis was applied in order to establish the extent to which the analyzed variables are able to predict a particular outcome. An analysis of the overall attitude towards tourism included two dependent variables: Y1 ("support for sustainable tourism") and Y2 ("overall opinion about tourism in the future"), as well as 24 independent variables (13 tourism impact variables, seven variables related to the control of tourism development and understanding of its impact on the community, four socio-demographic variables). Assessment of the model (in order to test multi-colinearity, the values of tolerance and the variance inflation factor were taken into account when analyzing the regression models) as a whole was achieved in this way.

According to the obtained results, several independent variables significantly predicted the degree of the community support for tourism in the Kopaonik National Park. However, the results showed that the part of variance of the dependent variable obtained in each of the regression models does not exceed the mean value ($r2$ is 0.34 for Y1 and 0.25 for Y2).

Table 8 shows the results of the first model. It can be concluded that four out of 24 independent variables made a statistically significant unique contribution, explaining 34% of the variance of the local population's support for tourism. Community members from the Kopaonik National Park who support the development of the tourism industry are marked by the following characteristics: they believe that the local population has easy access to places used by tourists; they believe that tourism does not encourage any rise in crime or decline in moral standards; and they support a larger presence of tourists in their community.

**Table 8.** Variables predicting the degree of support for sustainable tourism in the Kopaonik National Park.

| Variable | Standard. Coeff. Beta | Correlations | |
| --- | --- | --- | --- |
| | | Zero-Order | Part |
| Access to places used by tourists | 0.19 | 0.25 | 0.14 |
| Crime | 0.25 | 0.02 | 0.14 |
| Decline in moral standards | 0.19 | 0.07 | 0.13 |
| Attitude towards the presence of tourists in the community | 0.33 | 0.43 | 0.28 |

N = 195, $r^2$ = 0.342.

In the case of the second model, the results of a standard multiple regression analysis showed that three variables explained 25% of variance of the local population´s opinion about the development of tourism in the future. Members of the local community who believed that there should be more tourism in their surroundings in the future also believed that tourism created new jobs for the local population and improved activities of the local population. They also claimed that due to the tourism industry, the local population has easy access to places used by tourists (Table 9). As confirmed by the results, perceived socio-cultural and economic impacts have a strong influence on community support. Perceptions of the absence of negative effects (costs) and about legal and moral issues were also noticed. The literature explains that the economic benefits are often what is most valued by the community and easy to notice [32,34], and so the observed outcomes are somewhat expected. Regarding the absence of

negative effects, similar results were registered in previous studies on perceptions of communities in national parks [15].

**Table 9.** Variables predicting opinions about the development of tourism in the Kopaonik National Park in the future.

| Variable | Standard. Coeff. Beta | Correlations | |
| --- | --- | --- | --- |
| | | Zero-Order | Part |
| Creating new jobs for the local population | 0.25 | 0.20 | 0.17 |
| Activities of the local population | 0.17 | 0.15 | 0.14 |
| Access to places used by tourists | 0.18 | 0.17 | 0.13 |

### 4.3. Potential Bank Support

The results of the survey carried out on the sample of 21 commercial banks (26 banks operate in the system), which equals 80.77% of the total number of banks in the national banking system, give the following estimate in terms of the attitude of commercial banks towards lending in the field of tourism in the Kopaonik region.

The first group of questions deals with general attitudes regarding lending to the Kopaonik area, as well as the selection of the most appropriate activities for lending. The survey shows that the current banking practice has an affirmative stance about lending in the tourism sector in the Kopaonik region given that more than a half of respondents (63.16%) answered that they would finance different forms of tourist activities in this region.

When it comes to the attitudes towards the financing of economic activities in the Kopaonik region, respondents were given the question: "What would you most likely finance, on the condition that finances are satisfactory?" and the majority opted for the tourism industry (47.37%). A certain part of the sample would finance animal husbandry and agriculture (21.05%) and artisanship. A smaller number of the respondents (10.53%) answered that they would finance "other", i.e., the development of spa tourism and the economy through project financing. The survey showed that banks are not interested in financing forestry, hunting and fishing.

The next question in the survey referred to the type of project they would choose to finance in the Kopaonik region (on the condition that the project is viable and that the creditworthiness of the borrower is acceptable for their bank). The executives of the banks who participated in the survey provided different answers to this question; however, the majority opted for the financing of artisan-tourist activity (such as various private hospitality and accommodation facilities), (36.84%), while others opted for the construction of hotels (21.05%), entertainment parks and recreation centers (15.79%) and construction and improvement of the ski runs (10.53%). This type of choice was not expected given the high degree of profitability of the hotel industry. However, the financing of artisan-tourist activity presents a chance to involve the local population in the tourism industry. The respondents who answered they would finance other type of projects, i.e., 15.79% of them, would finance all types of infrastructure projects, the road network, projects relating to spa tourism, agriculture and fruit farming.

The specific part of the survey was related to the possible financing of the public enterprise "National Park Kopaonik". Although the respondents by a large percentage (68.42%) have a positive attitude towards this point, the majority of them (57.89%) see different restrictions in terms of financing the public enterprise given its ownership (ownership of the Republic of Serbia). Of those who indicated that restrictions existed, 27.27% pointed out that their business bank avoids the financing of public enterprises, while 72.73% specify restrictions reflected by the fact that enterprises owned by the Republic of Serbia are dependent on management structure, as well as legal restrictions and potential misuse. Consequently, the majority of respondents (78.95%) favor the financing of private companies

in the region compared to direct lending to the "National Park Kopaonik" public enterprise, due to the traditional non-flexibility of public enterprises and the state sector.

After carrying out the survey, interviews were conducted with ten executives of the banks to explain in detail the attitudes offered in the questionnaire. The results of the interviews indicate an inadequate degree of infrastructural development (the Kopaonik National Park and ski resort of Kopaonik have been devastated with unplanned construction and the absence of all forms of infrastructure—water supply, sewerage, road network, etc.). The majority of managers believe that, currently, accommodation capacities largely exceed the capacities of the ski resort. They see prospects in the financing of the construction of infrastructure, new ski runs and new different contents for potential tourists.

## 5. Discussion

A number of studies worldwide have examined the relationship between host communities and the development of tourism. The varied local responses mostly confirmed that benefits are the major impulse prompting people to support tourism and experience it positively [14,39,40]. As for predicting the level of community support, the results suggest that certain independent variables significantly predicted a specific outcome. Local support for the development of sustainable tourism was determined by specific positive perceptions regarding economic and sociocultural impacts of tourism. In addition, the attitudes of residents are also positive in regard to some non-economic variables, which has already been registered in some previous research [24]. All these results are in accordance with numerous studies that have noticed an important link between observed benefits to the host population and support for tourism activities [38,41,44,49], thereby supporting hypothesis H1 and suggesting that positive perceptions about various tourism impacts lead to support for the further development of sustainable tourism.

The study shows that socioeconomic variables affect the residents' perception to a great extent. The most significant variable affecting the opinions of residents is employment in the tourism industry. Community members employed in tourism had more favorable attitudes towards specific spheres of tourism impacts, and so the type of work was found to be a significant predictor of residents' perceptions about the effects of tourism (i.e., H3). The link between economic dependence and attitudes towards tourism has been pointed out by numerous researchers [61,62], and the results of our study are in agreement with some recent research highlighting the importance of this factor [59]. The results also confirm that younger and better educated people have more positive attitudes about the availability of proper information and correct understanding of the impact of tourism on the community. These categories of respondents strongly support the development of tourism, which can be explained by the fact that younger and better educated people have easier access to information in the era of digital media and are consequently familiar with the concept of sustainable tourism. Furthermore, as already pointed out, better educated people are prone to believe that there are potential job opportunities for them and benefits to be had from tourism, and so their support is partly expected. All of these findings appear to be in agreement with results of previous studies indicating that certain socioeconomic factors (place of residence, age, education) can noticeably influence the opinions of host residents [12,14,58]. Specifically, the results are consistent with previous research in national parks in Serbia [15] that emphasized the importance of socio-demographic characteristics for respondents' perceptions (i.e., H2).

Based on the results of this study, it can be stated that tourism has a minor influence on local benefits, and that the general population´s awareness of this industry is not at a satisfactory level. Inhabitants are not informed about measures for the control of tourism development, in addition to which money earned by tourism is not distributed transparently according to the local community. Although the community sees 'sustainable tourism' as positive, they are not aware of its meaning, nor do they know how to acquire knowledge about it. This is consistent with the results of some previous research exploring the benefits of tourism to local communities in protected areas and their

involvement in making decisions for the future development of tourism [27,58]. For an explanation, it is necessary to understand the broader picture of the development of settlements in recent decades. As previously mentioned, municipalities on the territory of the Kopaonik National Park belong to the category of underdeveloped municipalities in Serbia, ones with a level of development below the national average. This area is also characterized by a low gross domestic product and unemployment. Brankov et al. [15] explain that positive perceptions about tourism and its future development could be considerably affected by restriction of employment opportunities in other sectors of the local economy.

The conducted research shows that the positive attitudes of banks regarding lending in the field of tourism in the Kopaonik region could be used to raise the quality of life of the community and to activate them in tourism. The commercial banks are willing to finance different forms of tourist activities in this area (63% of confirmative answers), thus creating an opportunity for the potential economic recovery of the local residents. This would provide two types of benefits for the local community: employment in newly constructed facilities and the "push up" of private entrepreneurship. As the majority of banks opted for the financing of artisan-tourist activities (such as various hospitality and accommodation facilities), this should be used to encourage a small business in this sphere.

## 6. Conclusions

The main purpose of this study was to explore residents' perceptions about different impacts of tourism and specific factors that influence support for the tourism industry. An additional purpose of the research was to determine the attitudes of banks regarding potential lending that would involve the community.

Several conclusions can be drawn on the basis of results obtained in the present study. Generally, the surveyed population recognized various positive effects of tourism on the community's quality of life. An analysis of the questionnaire has identified positive thinking of the community about the tourism industry as a possible development force and positive attitudes towards this industry. Results of the research presented in this paper also identified a strong positive perception regarding the presence of visitors in the community, which is a good premise for future involvement of the local population in planning of tourism development. Various positive effects are highlighted, around which a majority consensus is achieved. Even though perceptions of positive impacts from tourism prevail, certain concerns about environmental issues on the Kopaonik Mountain were also manifested. According to the basic principles of SET, community members recognized that progress in tourism would produce more benefits than costs, and so they are willing to support its development.

The main theoretical contribution of this research is to stress the irreplaceable role that community residents perform in the development of tourism in rural areas facing long periods of recession and widespread underdevelopment. This study advances knowledge about predictors of support for the development of tourism, particularly in rural settings within protected areas. As the outcomes of specific relationships analyzed in this research represent original findings on national park-related community-based tourism, the present study also contributes significantly to the literature on this issue.

From the developmental and managerial points of view, different implications can be emphasized. Although tourism is interpreted as a catalyst that promotes socioeconomic and demographic development, it is clear that its strength depends on local factors—traffic conditions, geographical location, degree of urbanism, proximity to and importance of external centers of the population, the type of tourist center planned, demographic phenomena and processes transpiring in the concrete space of the site, etc. In the case of the Kopaonik National Park, it is important to resolve the gap that has arisen between the development of tourism on the one hand and community stagnation on the other. One of the conditions for increasing the quality of life in rural areas is the development and reconstruction of secondary tourist centers and settlements with tourist functions (primarily Jošanička Banja and Brzeće). This process should include the reconstruction of infrastructure, renewal of utilities equipment and reorganization of public services and public areas [82], which would reduce pressure on the tourist center in the highest part of the mountain and create additional employment of the

local population in tertiary activities (especially in tourism). Some authors suggest alternative forms for the development of tourism in this area, insufficiently promoted so far, such as geotourism [91] and health and recreational tourism [92]. An important prerequisite for communities in the national park and surrounding area to be included in tourism is the development and organization of local transportation infrastructure and recreational infrastructure. A significant number of rural settlements are characterized by poor traffic connections (unpaved roads or with poor-quality asphalt), with the tourist center in the highest part of Kopaonik Mountain, which is a limiting factor for tourists and overall development. This is also recognized by the banks that participated in the research and that are willing to finance various types of infrastructure projects.

It follows from the obtained results that providing proper financial incentives is an important step towards ensuring future community involvement in the tourism industry. In addition to direct sources of financing for the development of the national park (fees for use of the protected area at the disposal of the manager), emphasis should also be put on indirect sources, such as bank loans, various development funds, donations, etc. Besides the financial support to local residents, knowledge transfer is also necessary, since the majority of the population lack proper information about enterprises. That is why the public enterprise, as the national park´s managing body, together with local (and republic) management structures, should initiate the establishment of incentives and credit policy instruments in the wider area of the Kopaonik National Park. Investments and credit incentives should be especially focused on the development of agricultural production (improvement of livestock production), with simultaneous progress in tourism and complementary activities (improvement of tourist capacities, construction of communal infrastructure, etc.). Improved coordination of all participants in tourism (the park´s manager, tourist organizations, local self-governments, non-governmental organizations), with particular reference to stakeholder concept [93], is also necessary for the creation of specific guidelines and providing professional assistance to the community in different spheres of tourism management.

It should be emphasized at this point that the further validation of regression models and overall methodology in other regions is needed, since tourist destinations change in relation to the degree of the development of tourism and its impact. The present study can be interpreted as a recommendation to compare this mountain area with other tourist destinations that include similar features. However, it should be taken into account that the results are not generalizable, since specific local conditions (e.g., topography, culture, history) produce outcomes that, although they might have some characteristics in common with other destinations, are still unique to the particular area [94].

This research was carried out at a specific point in time and in particular conditions (underdevelopment of the wider region). As tourist destinations undergo transformation with the passage of time, this causes perceptions of the community about the impacts of tourism to change and evolve, and so future research should periodically analyze the relationship between residents' attitudes and destination modifications [44]. To provide a longitudinal approach to tourism development studies, it would be useful to carry out a follow-up study several years from now. The evolving nature of community perceptions could be investigated in that way.

This study has several limitations that should be addressed in future research. The primary focus of our study was on residents' perceptions about the impacts of tourism and models that explain their support for its development. Future research could consider additional predictors that affect community support for the development of tourism, factors such as the community attachment or place (destination) image [95]. Different factors that influence the attitude of residents towards tourism could also be included in a broader analysis (taking into account the length of residency, level of income, property ownership, etc.).

We point out that the present study is prevailingly quantitative in nature, tending to test the relationship between the variables that influence residents' attitudes towards tourism. As this type of research analyses what residents perceive, but does not necessarily explain why, Deery et al. [96] call for the use of qualitative studies, which would further enhance understanding and knowledge

about perceptions of the population in individual cases. In this particular case, a series of open-ended questions, as a specific supplementation of the questionnaire, could provide additional explanations.

Finally, our study is limited as a case study of a Serbian protected area. Various types of communities may harbor different opinions regarding the investigated problems. However, the obtained results can have broad importance for planners of tourism in areas where protected natural assets, rural communities of mountain areas and tourism come together. Further research should be particularly directed towards other regions of great importance due to their biodiversity and significant value as natural or heavily human-influenced landscapes for comparison with results of the present study.

**Author Contributions:** Conceptualization, J.B.; Methodology, J.B.; Formal analysis, J.B. and N.B.Ć.; Investigation, I.P. and B.Ž.; Data curation, I.P. and B.Ž.; Writing—original draft preparation, J.B., N.B.Ć. and B.Ž.; Writing—review and editing, J.B., N.B.Ć. and B.Ž.

**Funding:** This study was financially supported by the Ministry of Education, Science and Technological Development, Republic of Serbia (Grant No. III 47007).

**Acknowledgments:** We are grateful to members of the local community and executives of commercial banks for participation in the survey. We also thank Raymond Dooley for proofreading the final version of the text and editing the English.

**Conflicts of Interest:** The authors declare no conflict of interest.

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
