# Peer review of "Tourism as a Factor of Regional Development: Community Perceptions and Potential Bank Support in the Kopaonik National Park (Serbia)"

_sustainability, doi:10.3390/su11226507_

Round 1

Reviewer 1 Report

This article is well written and easy to read. I enjoyed reviewing this manuscript. There are major issues that should be corrected.

-Introductory sentences do not make any sense, and there is no coherence in the first paragraph.

-Line 39, second paragraph, the sentence is very long, immediately after author highlights the problem statements. I suggest authors to review critical literature, what has been done in the field of tourism and regional development and highlights its importance.

- The introduction is very long, I suggest to shorten it.

- The writing is sometimes a bit redundant, and in my opinion, some parts can be shortened and lightened, e.g. the first paragraphs of the introduction can be merged in a single one more

synthetic. In general, the introduction is interesting with much information. The introduction section includes a lot of information and sentences that are not well connected.

- The style of introduction must be coherent, and it should explain what the problem is, what has been researched in previous academic literature in this area, and what actually gap exists .further, how this study fulfils this research gap. I  suggest revising the whole introduction part, and provide what previously has been researched in this topic and what is the research gap?

- How the results of this study can be generalised to other companies.

- Separate the 2. Theoretical background and research hypothesis. Split it into two headings. Also the literature explains in the theoretical background does not depict actual theoretical background. I suggest authors to re-write theoretical section with appropriate use of theories for your literature.

- I have no problem with its use per se. However, throughout the paper, there is no attempt either to explain what it is (which could be easily done with some appropriate references), or, more importantly, to justify theory use. This would provide the relevant place to describe the theory (with relevant references) and to justify its use. This would best be done by reference to other relevant studies that have made use of this approach.

-Revise the hypothesis. It is not by the academic literature.

-Study Area heading 3 and heading 4 four should be combined and use as a subheading to explain the methodology.

- How potential items of the questionnaire should be framed from the previous studies, and it is quite important to dedicate a space on it in particular in the literature review. Then, you can use that review to show how you frame your questions in the methodology section. This reflects in the section of “composition of the questionnaire”. In particular, in my view, this part is twofold? 
1. how you measure the constructs with the “variables” from literature which should come in the literature review section. 
2. How you formulate the “questions” to measure the variables which come in the methodology section. 
I would strongly suggest differentiating because the former is expected much sooner in the paper. 

- Results analysis is very long, and it seems ambiguous. It can not get reader attention and use only correlational analysis to explain the results.

- I also think that the conclusion section could be strengthened by the provision of

more detail about the follow-on research outlined in the new paragraph.

I suggest authors may rewrite the conclusion part. It must start explaining the purpose and what has been done previously in the domain of studies. Also, provide future research directions precisely in the tourism innovation domain and suggest practical recommendations for the society and academics.

Author Response

Dear Reviewer#1,

The authors are very grateful for the time You devoted and for all the constructive comments that guided us through the making of all the necessary changes in manuscript structure. We have made serious improvements in accordance with all the comments and suggestions You recommended. The detailed responses are given below.

General comment of the reviewer:

This article is well written and easy to read. I enjoyed reviewing this manuscript. There are major issues that should be corrected.

Comment #1.1:

- Introductory sentences do not make any sense, and there is no coherence in the first paragraph.

Response to Comment #1.1:

The authors have deleted the first paragraph.

Comment #1.2:

-Line 39, second paragraph, the sentence is very long, immediately after the author highlights the problem statements. I suggest authors to review critical literature, what has been done in the field of tourism and regional development and highlights its importance.

Response to Comment #1.2:

The authors revised this sentence. We have also rewritten the whole Intoduction section, involving the new information on critical literature on residents’perceptions and deleting some general remarks. The other changes in the structure of this section are explained in the next comments.

Comment #1.3:

- The introduction is very long, I suggest to shorten it. The writing is sometimes a bit redundant, and in my opinion, some parts can be shortened and lightened, e.g. the first paragraphs of the introduction can be merged in a single one more synthetic. In general, the introduction is interesting with much information. The introduction section includes a lot of information and sentences that are not well connected.

Response to Comment #1.3:

As already mentioned, we have shortened the Introduction, by merging certain paragraphs and deleting some parts of the text.

Comment #1.4:

- The style of introduction must be coherent, and it should explain what the problem is, what has been researched in previous academic literature in this area, and what actually gap exists .further, how this study fulfils this research gap. I suggest revising the whole introduction part, and provide what previously has been researched in this topic and what is the research gap?

Response to Comment #1.4:

We deeply appreciate this comment. It helped us to properly structure the Introduction section. In the revised version of the paper, we provided an explanation and justification for our research, stating what research gaps are present and explaining how our research contributes to improving this situation.

Comment #1.5:

- How the results of this study can be generalised to other companies.

Response to Comment #1.5:

In a specific part of the Introduction, we gave an insight into how the results can be useful for different entities – participants in the tourism industry.

Comment #1.6:

- Separate the 2. Theoretical background and research hypothesis. Split it into two headings. Also the literature explains in the theoretical background does not depict actual theoretical background. I suggest authors to re-write theoretical section with appropriate use of theories for your literature. I have no problem with its use per se. However, throughout the paper, there is no attempt either to explain what it is (which could be easily done with some appropriate references), or, more importantly, to justify theory use. This would provide the relevant place to describe the theory (with relevant references) and to justify its use. This would best be done by reference to other relevant studies that have made use of this approach.

Response to Comment #1.6:

This comment was very useful to the authors. According to these comments, together with the instructions of other reviewers, we have completely rewritten the Theoretical background section. We have revised the hypotheses and formulated new ones that result from citing contemporary references dealing with the issues that are subject of our research. Since the majority of the reviewers suggested that hypotheses should be directly linked to the literature, we have decided not to separate theoretical background and research hypothesis into 2 headings, but to leave all the text as a whole.

Comment #1.7:

 -Revise the hypothesis. It is not by the academic literature.

Response to Comment #1.7:

As already mentioned in the previous response, we have revised the hypothesis and formulated the new ones, based on the literature review.

Comment #1.8:

-Study Area heading 3 and heading 4 four should be combined and use as a subheading to explain the methodology.

Response to Comment #1.8:

The authors have done this.

Comment #1.9:

- How potential items of the questionnaire should be framed from the previous studies, and it is quite important to dedicate a space on it in particular in the literature review. Then, you can use that review to show how you frame your questions in the methodology section. This reflects in the section of “composition of the questionnaire”. In particular, in my view, this part is twofold?

how you measure the constructs with the “variables” from literature which should come in the literature review section. How you formulate the “questions” to measure the variables which come in the methodology section.

I would strongly suggest differentiating because the former is expected much sooner in the paper. 

Response to Comment #1.9:

This was also very useful comment to the authors. It helped us to properly formulate the Theoretical background section. We gave an insight to triple-line approach that has been widely used in the tourism literature for explaining the different impacts of tourism to the community. We emphasized various types of impacts that are measured into each of the domains, which formed the basis for later editing of the questionnaire.

Comment #1.10:

- Results analysis is very long, and it seems ambiguous. It can not get reader attention and use only correlational analysis to explain the results.

Response to Comment #1.10:

The authors did not find a more logical way of presenting the results than the one already used. We emphasize that bivariate and multivariate statistical analysis were used to analyze the results as accepted techniques for interpreting the results. Also, part of the interpretation of the results involving the use of descriptive statistics has been shortened to some extent.

Comment #1.11:

- I also think that the conclusion section could be strengthened by the provision of more detail about the follow-on research outlined in the new paragraph. I suggest authors may rewrite the conclusion part. It must start explaining the purpose and what has been done previously in the domain of studies. Also, provide future research directions precisely in the tourism innovation domain and suggest practical recommendations for the society and academics.

Response to Comment #1.11:

The authors have rewritten the Conclusion section. Firstly, we have divided the Discussion and Conclusion section into two headings. In Discussion section we compared our results with previous studies, suggesting some matches or differences in the outcomes. The Conclusion section is supplemented by defining the purpose of the study as well as the theoretical contributions of the research. We have also pointed out in detail the future research directions and certain limitations of the study.

Reviewer 2 Report

The paper presents an interesting topic. It produces some interesting and useful results. In general terms the paper is well written and well argued. However, some revisions are needed.

1) Consider linking the hypotheses with the text it is based on (supplemented with sources).

2) The question is whether the number 3.6 in Table 2 (average for 13 variables) has a predicative ability when both positive and negative factors are included.

Author Response

Dear Reviewer#2,

We are very grateful for Your time to read and review our manuscript and for the constructive comments. All changes to the manuscript were made in accordance with the Reviewers’ comments and suggestions. The detailed responses are given below.

General comment of the reviewer:

The paper presents an interesting topic. It produces some interesting and useful results. In general terms the paper is well written and well argued. However, some revisions are needed.

Comment #2.1:

Consider linking the hypotheses with the text it is based on (supplemented with sources).

Response to Comment #2.1:

According to these comments, together with the instructions of other reviewers, we have completely rewritten the Theoretical background section. We have revised the hypotheses and formulated new ones that result from quoting contemporary references dealing with the issues that are subject to our research.

Comment #2.2:

The question is whether the number 3.6 in Table 2 (average for 13 variables) has a predictive ability when both positive and negative factors are included.

Response to Comment #2.2:

This is a very useful observation. We emphasize that this variable was not used in any of the analyzes as a standalone, for the reasons given by the reviewer. It is exclusively listed as the average value of all the 13 variables.

Reviewer 3 Report

This study is interesting example of presenting local problems in tourism development in Serbia for wider public. The conceptual and methodological dimensions of the paper are correct. The results of this study are supported by survey instrument based on the local questionnaire model recommended by the UN World Tourism Organization. The paper is well organised, coherent and clearly written. However, I would like to make a few observations which I believe may contribute to improving the clarity of the study:

Firstly, authors should broaden the theoretical part of the article. Secondly, combining two such different and separate issues as the attitudes of the local community towards the tourism and bank support for the development of tourism in one article is inappropriate and unnecessary. A commercial bank's approach to lending in the field of tourism is a good research topic for the next article. Thirdly, the particular stream of literature that this paper aims to contribute to is not clearly identified. I suggest to improve the theoretical framework. It is very important to place your study in relevant scientific fields. It is not clear how this paper contributes to the extant literature. Further work is needed in this regard. What are the scientific fields that this research contributes to? What is the major contribution of this paper into knowledge? This point needs to be more clearly stated in the paper and conclusions. Fourth, the conclusions should emphasize the theoretical implications of study more strongly, indicate the research limitations and suggest also directions for further research.

Author Response

Dear Reviewer#3,

The authors deeply appreciate Your time spent reading and reviewing this manuscript. We thank you for all the constructive comments and suggestions. All changes to the manuscript were made in accordance with the Reviewers’ comments and suggestions. We gave detailed responses below.

General comment of the reviewer:

This study is interesting example of presenting local problems in tourism development in Serbia for wider public. The conceptual and methodological dimensions of the paper are correct. The results of this study are supported by survey instrument based on the local questionnaire model recommended by the UN World Tourism Organization. The paper is well organised, coherent and clearly written. However, I would like to make a few observations which I believe may contribute to improving the clarity of the study:

Comment #3.1:

Firstly, authors should broaden the theoretical part of the article.

Response to Comment #3.1:

The authors have done this. The ways in which this has been done are described in the comments below.

Comment #3.2:

Secondly, combining two such different and separate issues as the attitudes of the local community towards tourism and bank support for the development of tourism in one article is inappropriate and unnecessary. A commercial bank's approach to lending in the field of tourism is a good research topic for the next article.

Response to Comment #3.2:

Here we have an obligation to provide an explanation and justification for the inclusion of a part of research related to the attitudes of the banks into this study. Firstly, we didn’t want our research to be finished after the analyses of the perceptions. After this analysis, the question is how to proceed and how to solve the problems identified. The second part of the research gives an important and realistic answer to these questions.

Analysis of the questionnaire has identified positive thinking of the community about the tourism industry as a possible development force, but this isn’t enough. One of the highlighted problems, when it comes to all communities in protected areas in Serbia, is the lack of start-up capital among the local population, so for these stakeholders, the eventual bank lending in the sphere of the small-business could be crucial. This is why we included this specific dimension in our research. This allowed us to point out future directions of action, i.e. to offer concrete solutions.

However, we deeply agree that this issue can be investigated in more detail as a separate article and we appreciate this suggestion very much.

Comment #3.3:

Thirdly, the particular stream of literature that this paper aims to contribute to is not clearly identified. I suggest to improve the theoretical framework. It is very important to place your study in relevant scientific fields. It is not clear how this paper contributes to the extant literature. Further work is needed in this regard.

Response to Comment #3.3:

Thank you very much for this comment. According to instructions of all of the reviewers, we have completely rewritten the Theoretical background section. We have pointed out in more detail to the social exchange theory and the triple-line approach that conceptually framed this study. We have also revised the hypotheses that are now directly related to the literature review.

Comment #3.4:

What are the scientific fields that this research contributes to? What is the major contribution of this paper into knowledge? This point needs to be more clearly stated in the paper and conclusions. Fourth, the conclusions should emphasize the theoretical implications of study more strongly, indicate the research limitations and suggest also directions for further research. 

Response to Comment #3.4:

This was a very useful direction for the authors to properly improve the Conclusion section. Firstly, the purpose of the study, as well as the main contributions of the research have been highlighted in this section. Secondly, we gave an insight into the specific theoretical implications of the study. Thirdly, we have also pointed out in detail the future research directions and certain limitations of this study.

Reviewer 4 Report

1) As “there are few official documents” reporting and detailing the role of “local residents in the planning and management of tourism in their communities”, a “methodology relying on surveys (based on the “social exchange theory (SET)”), with the aim of identifying perceptions and specific factors that influence support for the tourism industry”, was applied by the authors to “improve the situation, clarify local attitudes and determine the attractiveness of the area for potential lending by banks”.

2) The “Introduction” section seems to be comprehensive enough as it presents with “acceptable detail and extent” the rationale behind their paper.

3) The section on “Theoretical background and research hypothesis” is welcome as it adds academic and pedagogical values to the paper.

4) Their methodological approach looks like “a tailored, purpose-oriented one” as “a survey instrument” was specifically designed and implemented by the authors to achieve the goals they defined for the research reported in the paper.

5) However, despite the large “Statistical and Data Analysis” performed by the authors along the paper and the references 26, 36, 64, 71, 72, 81 and 87, I have not been able to find any REFERENCES to any “Statistical and Data Analysis-dedicated” books or papers.

6) Although the “Discussion and conclusions” are, as expected, based on previous works performed by others and the authors themselves, they seem, however, to rely “heavily”? on a particular paper of the authors: see for instance, reference 21.

7) Finally, in addition to the comments that were made above, some minor spell check, formatting and wording issues such as those listed below are still needed or should be addressed by the authors:

pg 5, line 187, while the authors write that “only” “three research hypotheses were developed”, “four” hypotheses (H1, H2, H3 and H4) are, however, reported on lines 188 -192; this should be checked; pg 11, Table 3 and pg 12, lines 395- 401: the “code” assigned to each variable in Table 3, pg 11 and those referred to in the text, pg 12, lines 395- 401 do not match (see, for instance, the case of variable V2)? This should also be checked; pg 14, lines 462-464 and pg 15, line 474: the number of “independent variables” analysed in this research should be checked: were they 23 or 24?? pg 5, line 204 and pg 7, line 252, every abbreviation (see for instance, a.s.l. (line 204) and “above sea level (line 252) and other cases in the paper) should be defined at its first occurrence in the paper/text.

Author Response

Dear Reviewer#4,

The authors deeply appreciate Your time spent reading and reviewing this manuscript. We are very grateful for all the constructive comments that helped us to significantly improve our research. All changes to the manuscript were made in accordance with the Reviewers’ comments and suggestions. We gave detailed responses below.

General comment of the reviewer:

As “there are few official documents” reporting and detailing the role of “local residents in the planning and management of tourism in their communities”, a “methodology relying on surveys (based on the “social exchange theory (SET)”), with the aim of identifying perceptions and specific factors that influence support for the tourism industry”, was applied by the authors to “improve the situation, clarify local attitudes and determine the attractiveness of the area for potential lending by banks”.

Comment #4.1:

The “Introduction” section seems to be comprehensive enough as it presents with “acceptable detail and extent” the rationale behind their paper.

Response to Comment #4.1:

We additionally provided an explanation and justification for our research in the Introduction section. We believe that this has helped to improve the Introduction to a significant extent.

Comment #4.2:

The section on “Theoretical background and research hypothesis” is welcome as it adds academic and pedagogical values to the paper.

Response to Comment #4.2:

The authors appreciate this comment. However, according to the instructions of other reviewers, it was necessary for us to completely rewrite the Theoretical background section. We have revised the hypotheses and formulated new ones that result from citing contemporary references dealing with the issues that are subject of our research.

Comment #4.3:

Their methodological approach looks like “a tailored, purpose-oriented one” as “a survey instrument” was specifically designed and implemented by the authors to achieve the goals they defined for the research reported in the paper.

Response to Comment #4.3:

We emphasize that the survey instrument used in this research was not designed by the authors. We have pointed out in the manuscript that the local questionnaire model was recommended by the UN World Tourism Organization. The authors performed categorization of the existing variables using the triple bottom line approach to impacts, commonly employed in studies dealing with the sustainable development of tourism. After that, each of the domains of tourism impact has been analysed.

Since the adopted local questionnaire model analyzes the benefits and costs of the tourism (favourable and unfavourable effects), it was logical for the authors to accept the SET theory as the theoretical framework for this study. Another advantage of using this theory was the higher predictive strength of the models.

Comment #4.4:

However, despite the large “Statistical and Data Analysis” performed by the authors along the paper and the references 26, 36, 64, 71, 72, 81 and 87, I have not been able to find any REFERENCES to any “Statistical and Data Analysis-dedicated” books or papers.

Response to Comment #4.4:

This was a very useful suggestion for the authors. All performed statistical analysis was certainly made on the basis of consulting the specific literature that provided a background knowledge in this area. We have corrected oversight pointed out by the reviewer and we have cited this literature in the revised version of the manuscript.

Comment #4.5:

Although the “Discussion and conclusions” are, as expected, based on previous works performed by others and the authors themselves, they seem, however, to rely “heavily”? on a particular paper of the authors: see for instance, reference 21.

Response to Comment #4.5:

The reference 21 (in the revised version of the manuscript it is the reference 15) is actually the research of the authors that have been performed in other national parks in Serbia recently, using the same methodology as it has been the case in this study. As the sampling methods, data collection tools and data analysis techniques were identical, it provided a realistic basis for meaningful comparison. This was the main reason why the results obtained in this research have been compared to a great extent with this particular study.

In order to adopt the suggestion from the reviewer, we have partially rewritten the Discussion section and compared our results with larger number of previous studies, suggesting some matches or differences in the outcomes.

Comment #4.6:

Finally, in addition to the comments that were made above, some minor spell check, formatting and wording issues such as those listed below are still needed or should be addressed by the authors:

pg 5, line 187, while the authors write that “only” “three research hypotheses were developed”, “four” hypotheses (H1, H2, H3 and H4) are, however, reported on lines 188 -192; this should be checked; pg 11, Table 3 and pg 12, lines 395- 401: the “code” assigned to each variable in Table 3, pg 11 and those referred to in the text, pg 12, lines 395- 401 do not match (see, for instance, the case of variable V2)? This should also be checked; pg 14, lines 462-464 and pg 15, line 474: the number of “independent variables” analysed in this research should be checked: were they 23 or 24?? pg 5, line 204 and pg 7, line 252, every abbreviation (see for instance, a.s.l. (line 204) and “above sea level (line 252) and other cases in the paper) should be defined at its first occurrence in the paper/text. 

Response to Comment #4.6:

The authors have corrected all the listed inconsistencies in the text.

Reviewer 5 Report

This manuscript should be submitted to a professional proof reader. 

The abstract suggested that the research involved a qualitative case study and a survey questionnaire (i.e. quantitative study). At this stage, it is unclear whether the researchers have used valid measures in their study. The findings and the implications of this study were somewhat expected. The authors should elaborate further on the implications of study. They need to explain how their paper adds value to this journal. They need to provide more details.

The introduction has included some rhetoric on the relationship between tourism, the local community and the environment. The authors did not explain the justification / rationale of their study. I think that they need to clarify the research question of this contribution. What exactly are they researching?

The literature review should be drawn from recent, high impact journals. The theoretical underpinnings should lead to the hypotheses. The researchers should build their hypotheses on the theoretical foundations. The hypotheses are not linked with the relevant literature.

The researchers are promoting the Serbian park and its outstanding beauty. They would like to raise awareness about its existence with the readers of this journal. They provide a detailed description of their tourist attraction. Well, the sustainability of this park seems to be very similar to the parks that are located in other regions and territories, as the authors discuss about the usage of the rural land and of the mountain areas for tourism and recreation, etc. 

The authors suggested that they gathered their data via a face to face survey questionnaire. They reported that their questions were based on UNWTO measures. Therefore, the researchers could not confirm the validity and reliability of the the measures as they were not tried and tested in academia. Yet, they reported that the respondents were asked about "economic", "socio cultural", "environmental" impacts of tourism, "legal and moral issues", etc.   

Afterwards, the researchers reported that they analysed the "attitudes of commercial banks". In this case they submitted their survey questionnaire via email. I can't understand why the researchers gathered more data from the banks. This part of this study could be a distinct research that could possibly be presented in another paper. The authors have conducted two different studies in this paper.

The researchers have presented their descriptive statistics and discussed about the findings. This was not a difficult task. However, the standard of this journal is (should be) much higher than this descriptive research. 

Regrettably, the findings of this study were somewhat expected. The results from this research has often been reiterated in various papers in academia. There is nothing novel here. Moreover, I believe that the implications of this study are not adding value to this journal. the researchers need to clarify how their contribution is different than other papers on sustainable tourism.

I hope that you will improve your paper before submitting it to another journal. Best wishes.

Author Response

Dear Reviewer#5,

The authors deeply appreciate Your time spent reading and reviewing this manuscript. We thank you for all the constructive comments and suggestions. All changes to the manuscript were made in accordance with the Reviewers’ comments and suggestions. We gave detailed responses below.

General comment of the reviewer:

The abstract suggested that the research involved a qualitative case study and a survey questionnaire (i.e. quantitative study). At this stage, it is unclear whether the researchers have used valid measures in their study. The findings and the implications of this study were somewhat expected. The authors should elaborate further on the implications of study. They need to explain how their paper adds value to this journal. They need to provide more details.

Comment #5.1:

The introduction has included some rhetoric on the relationship between tourism, the local community and the environment. The authors did not explain the justification / rationale of their study. I think that they need to clarify the research question of this contribution. What exactly are they researching?

Response to Comment #5.1:

According to these comments, together with the instructions of other reviewers, we have improved the Introduction section. In the revised version of the paper, we provided an explanation and justification for our research by stating which research gaps are present and by explaining how our research contributes to improvement of the situation.

Comment #5.2:

The literature review should be drawn from recent, high impact journals. The theoretical underpinnings should lead to the hypotheses. The researchers should build their hypotheses on the theoretical foundations. The hypotheses are not linked with the relevant literature.

Response to Comment #5.2:

We appreciate this comment very much. According to these instructions, we have completely rewritten the Theoretical background section. We have revised the hypotheses and formulated new ones resulting from the contemporary literature dealing with the issues that are subject of our research.

Comment #5.3:

The researchers are promoting the Serbian park and its outstanding beauty. They would like to raise awareness about its existence with the readers of this journal. They provide a detailed description of their tourist attraction. Well, the sustainability of this park seems to be very similar to the parks that are located in other regions and territories, as the authors discuss the usage of the rural land and of the mountain areas for tourism and recreation, etc. 

Response to Comment #5.3:

We can argue on this issue since the specific local conditions that characterize this Park are not present in other regions and territories.

We emphasized that conflicts between tourism and active protection in this Park have grown over time, since the highest parts of the mountain represent the largest ski resort in the country and, at the same time, a protected natural area. We also pointed out to a large number of entities that participate in organization and presentation of the tourist offer. In such circumstances, the achievement of community well-being may be the greatest challenge, but also a necessary condition to acquire sustainability of the wider territory.

Comment #5.4:

The authors suggested that they gathered their data via a face to face survey questionnaire. They reported that their questions were based on UNWTO measures. Therefore, the researchers could not confirm the validity and reliability of the measures as they were not tried and tested in academia. Yet, they reported that the respondents were asked about "economic", "socio cultural", "environmental" impacts of tourism, "legal and moral issues", etc.   

Response to Comment #5.4:

Even though there are certainly realistic grounds for this concern, we would like to point out that validity and reliability of every research scale depend on the specific sample. This is especially important for reliability which in this particular study has been tested at the full-scale level and additionally at the subscale level. As Cronbach’s alpha coefficient is sensitive to short scales, we calculated the mean inter-item correlation between the items of all subscales and the range of results was within the recommended values.

Comment #5.5:

Afterwards, the researchers reported that they analysed the "attitudes of commercial banks". In this case, they submitted their survey questionnaire via email. I can't understand why the researchers gathered more data from the banks. This part of this study could be a distinct research that could possibly be presented in another paper. The authors have conducted two different studies in this paper.

Response to Comment #5.5:

Here we are obligated to provide an explanation and justification for the inclusion of a part of research related to the attitudes of the banks into the study. Firstly, we didn’t want our research to be finished after the analyses of the perceptions. After this analysis, the question is how to proceed and how to solve the problems identified. The second part of the research gives an important and realistic answer to these questions.

Analysis of the questionnaire has identified positive thinking of the community about the tourism industry as a possible development force, but this wasn’t enough. One of the highlighted problems, when it comes to all communities in protected areas in Serbia, is the lack of start-up capital among the local population, so for this stakeholders, the eventual bank lending in the sphere of the small-business could be crucial. This is why we included this specific dimension in our research. This allowed us to point to future directions of action, i.e. to offer concrete solutions. What is more important, we included only the executives in our research as their opinion is most representative and carries the greatest “weight”.

However, we deeply agree that this issue can be investigated in more detail as a separate article and we appreciate this suggestion very much.

Comment #5.6:

The researchers have presented their descriptive statistics and discussed about the findings. This was not a difficult task. However, the standard of this journal is (should be) much higher than this descriptive research. 

Response to Comment #5.6:

The authors can not agree with this statement. Indeed, one part of our study involved the use of descriptive statistics to analyze the results. However, our research also included bivariate (t-tests, ANOVA) and multivariate (multiple regression) statistical analysis as accepted research techniques. Different regression models for predicting the level of community support were also created by performing the multiple regression analysis.

Comment #5.7:

Regrettably, the findings of this study were somewhat expected. The results from this research has often been reiterated in various papers in academia. There is nothing novel here. Moreover, I believe that the implications of this study are not adding value to this journal. The researchers need to clarify how their contribution is different than other papers on sustainable tourism.

Response to Comment #5.7:

Although the results obtained may have similarities with other paper investigating the same subject, specific local conditions (in this case the second largest ski centre in Southeast Europe on one side and the extremely underdeveloped communities on another side) produce outcomes that are still unique to this particular area. Through the research, we have highlighted the specific difficulties that communities face and outlined possible solutions for improving the quality of life that would include tourism. The results obtained by investigating the attitudes of the banks are unique and allow us to open up a whole new dimension in exploring the potential benefits of local communities from tourism in protected areas.

Finally, we consider the main theoretical contribution of this research is to point out the irreplaceable role that community residents perform in the potential development of tourism in rural areas facing long periods of recession and widespread underdevelopment.

Round 2

Reviewer 1 Report

Paper has now been improved significanlty. One minor thing, i suggest to remove Table.10.

Author Response

Dear Reviewer,

The authors deeply appreciate your additional time spent reading and reviewing this manuscript. We acted according to the suggestions and the detailed responses are given below:

Comment #1.1:

Paper has now been improved significantly. One minor thing, I suggest to remove Table.10.

Response to Comment #1.1:

The authors have removed Table 10. from the manuscript. Given that it contained some of the research findings, the text of section 4.3. Potential bank support was modified to a lesser extent.

Reviewer 5 Report

The authors have improved many areas in the manuscript. I suggested in my previous review that the authors should have focused on one study, rather than presenting two distinct studies in one paper. The second study among banks is not adding value to this paper. This study is not related with the 1st study. It is not even complementing it.

Moreover, I am concerned on the measures that were drawn from the UNWTO. I reiterate that these measures were not academic. Therefore, they are not appropriate for an empirical study that is submitted to a highly indexed journal (like MDPI's Sustainability). The measuring items that were used in your study were not reliable (and could never be), even though you reported your cronbach's alpha.  No other researcher could replicate your study as your measures are flawed. 

For these reasons, I believe that the standard of this paper is very low. The authors are not presenting a novel contribution. The findings in this study have already been reported in other papers.

I am aware that this is a very critical review, however, I believe that you can do another research on this topic. You can use appropriate measures for your new study. If you do so, there are other analytical techniques that can be done with your dataset (E.g. factor analysis, SEM, etc). I do not want to discourage you from submitting a study on this Serbian park (you can still use different parts of this paper's literature review). However, your methodology and analysis are very weak at this stage. 

Best wishes,

Author Response

Dear Reviewer#5,

The authors are grateful for all the comments and suggestions that are given in your review. We deeply respect all the different academic opinions. Our explanations follow:

Comment #5.1:

The authors have improved many areas in the manuscript. I suggested in my previous review that the authors should have focused on one study, rather than presenting two distinct studies in one paper. The second study among banks is not adding value to this paper. This study is not related with the 1st study. It is not even complementing it.

Response to Comment #5.1:

Although it is a smaller part of our research, the analysis of banks' attitudes is very important for determining the real opportunities that the community have for engaging in tourism in Kopaonik area. We consider this part of research to be an extension of an analysis of community perceptions that gives an important direction for the future acting of community members in the sphere of private entrepreneurship in the tourism industry. As such, it is an important complement to our main research.

Comment #5.2:

Moreover, I am concerned on the measures that were drawn from the UNWTO. I reiterate that these measures were not academic. Therefore, they are not appropriate for an empirical study that is submitted to a highly indexed journal (like MDPI's Sustainability). The measuring items that were used in your study were not reliable (and could never be), even though you reported your cronbach's alpha.  No other researcher could replicate your study as your measures are flawed. For these reasons, I believe that the standard of this paper is very low. The authors are not presenting a novel contribution. The findings in this study have already been reported in other papers.

Response to Comment #5.2:

Although we deeply appreciate the reviewer’s opinion, we can not agree with this statement. There are specific arguments that we are obliged to make in order to support the previous claim.

In our research, we have used Local Questionnaire Model recommended by the UNWTO. The experts of the UNWTO, while explaining this model, emphasize that “several model questions are provided addressing many of the issues found to be key in past studies and applications” (UNWTO, 2004). This is supported by the fact that measures used in this questionnaire model can be found in various tourism literature using the triple bottom line approach to impacts. It is also emphasized that this approach is consistent with recommended practice for questionnaires of this type. This allows different researchers to repeat the research using the same methodology within a different study area.

Further, as it has already been pointed out, we additionally checked the reliability and validity of the measures on our sample and the range of results were within the recommended values.

Based on all of the above, there is no realistic basis for claiming that the measures used in this research are not reliable or academic.

Comment #5.3:

I am aware that this is a very critical review, however, I believe that you can do another research on this topic. You can use appropriate measures for your new study. If you do so, there are other analytical techniques that can be done with your dataset (E.g. factor analysis, SEM, etc). I do not want to discourage you from submitting a study on this Serbian park (you can still use different parts of this paper's literature review). However, your methodology and analysis are very weak at this stage. 

Response to Comment #5.3:

Thank you very much for your opinion and recommendations. The authors gave the arguments and explained their different opinion in the previous statement.